# High-resolution light field prints by nanoscale 3D printing

John You En Chan [1], Qifeng Ruan [1✉], Menghua Jiang[2], Hongtao Wang [1], Hao Wang [1], Wang Zhang [1], Cheng-Wei Qiu [2] & Joel K. W. Yang [1,3✉]

A light field print (LFP) displays three-dimensional (3D) information to the naked-eye observer under ambient white light illumination. Changing perspectives of a 3D image are seen by the observer from varying angles. However, LFPs appear pixelated due to limited resolution and misalignment between their lenses and colour pixels. A promising solution to create high-resolution LFPs is through the use of advanced nanofabrication techniques. Here, we use two-photon polymerization lithography as a one-step nanoscale 3D printer to directly fabricate LFPs out of transparent resin. This approach produces simultaneously high spatial resolution (29–45 μm) and high angular resolution (~1.6°) images with smooth motion parallax across 15 × 15 views. Notably, the smallest colour pixel consists of only a single nanopillar (~300 nm diameter). Our LFP signifies a step towards hyper-realistic 3D images that can be applied in print media and security tags for high-value goods.

[1] Engineering Product Development, Singapore University of Technology and Design, Singapore, Singapore. [2] Department of Electrical and Computer Engineering, National University of Singapore, Singapore, Singapore. [3] Institute of Materials Research and Engineering, Singapore, Singapore. ✉email: qifeng_ruan@sutd.edu.sg; joel_yang@sutd.edu.sg

Commonly available prints have a fixed two-dimensional (2D) appearance as they store only intensity and colour information. Unfortunately, these prints lack a vital piece of information—the directional control of light rays[1–4]—to display a three-dimensional (3D) image. With no ability to discriminate the direction of light rays, the prints appear unchanged to an observer from all viewing angles. On the other hand, light field prints (LFPs) encode directional information, which enables them to display changing perspectives of a 3D image seen from varying viewing angles. This technique of displaying a 3D image was discovered in 1908 by the Nobel Laureate, Gabriel Lippmann[5]. He proposed using an array of tiny lenses on film to record a scene, so that each lens formed its own sub-image with a slightly shifted perspective. The film would then be illuminated under diffuse light to reconstruct an integral 3D image of the scene. LFPs are autostereoscopic as their 3D images are visible to the naked eye under incoherent and unpolarized illumination, which gives them an advantage over stereoscopic prints[6,7] that require viewing through a pair of anaglyph glasses or orthogonally polarized filters. LFPs also do not require laser illumination used in conventional holograms[8]. However, LFPs appear pixelated because of low spatial resolution and low angular resolution caused by fabrication limitations. Spatial resolution is limited by the centre-to-centre separation between lenses, and angular resolution is limited by the density of pixels under each lens[9,10].

To create high-resolution LFPs, we leverage on advanced nanofabrication techniques in structural colour printing[11–24]. Structural colour prints with plasmonic nanostructures[16,18] can attain a pixel resolution of ~100,000 dots per inch, which is several orders of magnitude higher than that of conventional inkjet prints[17]. Though inkjet printers are capable of printing LFPs for art[25], their pixel resolution is insufficient for optical security devices[26] that require the spatial resolution to be <50 μm (i.e. beyond the resolving power of human vision). Moreover, misregistration between colour layers is an issue in these printers, but not in nanofabrication tools, such as electron beam lithography that position plasmonic colour pixels without misregistration[14]. Jiang et al. demonstrated high-resolution multicolour motion effects by bonding a microlens array onto a plasmonic colour print[27]. However, this approach is relatively complicated because it requires multiple processing steps and manual alignment between the microlenses and pixels. Accurate alignment is essential to reconstruct clear 3D images but doing manual alignment at the nanoscale is challenging as it relies on delicate movements and visual inspection. As manual alignment has been done for regular arrays of microlenses and colour pixels, an alternative alignment approach is thus needed for irregular arrays that have a pseudorandom arrangement[28].

In this work, we use two-photon polymerization lithography (TPL) to fabricate high-resolution LFPs in one patterning step that circumvents the need for doing manual alignment. The microlenses and structural colour pixels of our LFPs are aligned automatically in the TPL system (Nanoscribe GmbH Photonic Professional GT system), which can position each volumetric pixel exposed by the laser up to an accuracy of 10 nm. As TPL is an additive manufacturing technology, we fabricate the microlenses and structural colour pixels in discrete slicing height steps of 20 and 300 nm, respectively. The microlenses and structural colour pixels are made of the same low refractive index material, IP-Dip photoresist (n ~1.55). Unlike plasmonic colour pixels, our structural colour pixels do not require additional metal deposition, which makes the TPL system only necessary for fabricating LFPs. The microlenses and structural colour pixels are fabricated together in a pseudorandom arrangement, which can minimize undesired moiré patterns and also encode secret information for security applications. More importantly, our LFP displays simultaneously high spatial resolution (29–45 μm) and high angular resolution (~1.6°) images with smooth motion parallax that appear unpixellated to the naked eye, even up close.

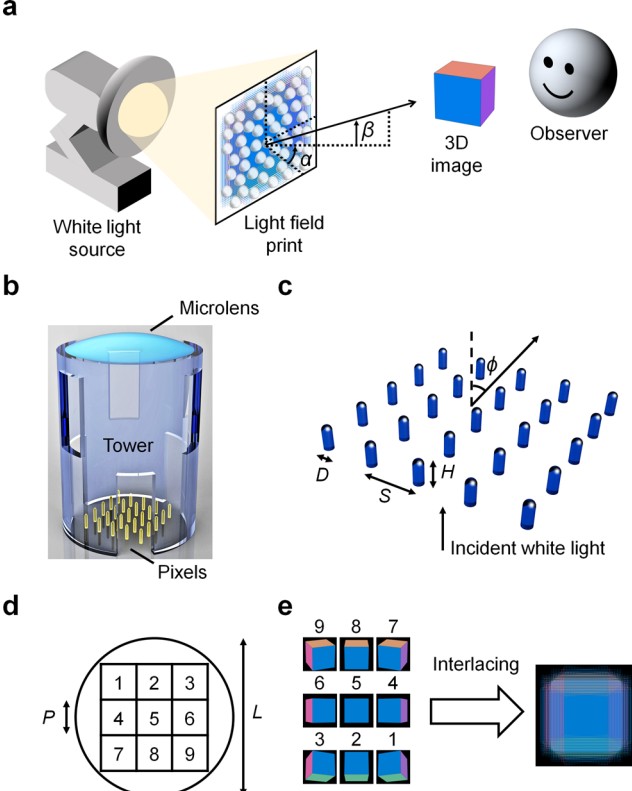

**Fig. 1 Design of the light field print. a** Schematic that illustrates the working principle of our light field print (LFP). A white light source is used to illuminate the structural colour pixels in the LFP. Transmitted light from the pixels is collected by the microlenses and projected to the far field. An observer in the far field sees a colourful 3D image from the viewpoint denoted by azimuthal angle α and elevation angle β. **b** Schematic of one display unit: A tower supports a spherical plano-convex microlens placed one focal length above a block of structural colour pixels that contains an array of nanopillars. **c** Schematic of one structural colour pixel that contains 5 × 5 nanopillars, in which each nanopillar has diameter D, height H and pitch S. ϕ is the viewing angle of the pixel. **d** Plan-view schematic of 3 × 3 pixels in a display unit. L is the diameter of the microlens, represented by a circle; P is the pitch of each pixel, represented by a square. Each pixel is assigned a number 1–9 and it is extracted from an input image assigned with the same number. **e** The input images are interlaced to generate a digital map of pixel positions for printing the LFP.

## Results

**Design concepts.** We propose the working principle of our LFP: A white light source is used to illuminate the structural colour pixels, which function as colour filters that transmit varying intensities of visible wavelengths within a range of angles collected by the microlenses. The microlenses then project the collected light to an observer who sees the colours of selected pixels at the corresponding viewpoint in the far field. All these selected pixels form a complete 3D image (Fig. 1a). The design of our LFP was inspired by Thiele et al.'s compound microlens system[29]. Instead of a compound system, we designed our system (Fig. 1b) with a hollow tower to support a spherical plano-convex microlens above a block of structural colour pixels that contains an

array of nanopillars[8,30,31]. We refer to this system as a display unit. Our LFP comprised $65 \times 65$ display units placed in a pseudorandom arrangement with centre-to-centre separation between 29 and 45 μm. A centre-to-centre separation <50 μm ensured that the 3D image appeared unpixellated to the naked eye. In each display unit, the tower was constructed with several openings to allow uncured photoresist to be washed away after printing. The tower had height = 34.6 μm and diameter = 26 μm, whereas the microlens had radius of curvature $R = 22$ μm and diameter $L = 21$ μm. Each microlens was placed one focal length ~37 μm above the structural colour pixels to obtain a focused image (see "Raytracing simulations" section). The diameter and focal length of our microlens corresponded to a numerical aperture (NA) of ~0.28, designed to cover an acceptable range of viewing angles ($\phi = 0$–16°) that was limited by our structural colour pixels (see "Electromagnetic wave simulations" section). This NA yielded a diffraction-limited focal spot size of ~1 μm, which equalled the smallest pixel pitch $P$ in our design. Initially, we designed $3 \times 3$ pixels per display unit. Each pixel comprised $5 \times 5$ nanopillars ($P = 5$ μm) with diameter $D$, height $H$, and pitch $S$ (Fig. 1c). Subsequently, we increased the number of pixels per display unit to $5 \times 5$ pixels ($P = 3$ μm, $3 \times 3$ nanopillars per pixel) and $15 \times 15$ pixels ($P = 1$ μm, single nanopillar per pixel). In every display unit, the block of pixels fits within the circular cross-section of the microlens (Fig. 1d). The number assigned to each square represents a pixel extracted from an input image with that number, e.g. a pixel labelled '1' is extracted from the first input image. Hence, the number of pixels per display unit equals the number of input images. We generated nine input images of a multi-coloured cube with different perspectives, and used an algorithm[32] to create an interlaced digital image that maps the pixel positions to their display units (Fig. 1e). Note that the order of numbers assigned to the input images is flipped horizontally and vertically from the order of numbers assigned to the pixels (equivalent to a 180° rotation about the optical axis). This order ensured that the LFP displayed a 3D image with proper depth and motion parallax, otherwise it would appear pseudoscopic (i.e. depth-inverted)[9,33]. In the design process, we performed raytracing simulations to determine the microlens focal length and acceptable range of viewing angles. We also performed full electromagnetic wave simulations to determine the microlens focal spot size and the colour change of pixels with viewing angle.

**Raytracing simulations.** As our structural colour pixels had a limited acceptable range of viewing angles ($\phi = 0$–16°), they required microlenses with NA ~0.28. To fulfil this requirement, we designed spherical plano-convex microlenses with diameter $L = 21$ μm and radius of curvature $R = 22$ μm. From the lens-maker equation, we derived the theoretical focal length of our microlenses to be $F' = 40$ μm for refractive index $n = 1.55$. The theoretical focal length is expressed in Eq. 1:

$$F' = \frac{R}{n-1} \qquad (1)$$

To validate our design, we simulated parallel incident light rays on the convex side of the microlens for field angle $\theta = 0°$ and sampling wavelengths $\lambda = \{450, 532, 635\}$ nm (Supplementary Note 1). We refer to field angle as the angle between the incident ray and optical axis. For the respective wavelengths, we set $n = \{1.56, 1.55, 1.54\}$ according to the Cauchy parameters of UV cured IP-Dip photoresist[34]. After accounting for aberrations, we found that the focal plane of the microlens was located at $\sim z = 37$ μm below its vertex (Fig. 2a). We used this value as our design focal length $F$, which is in close agreement with $F'$. Based on the principle of reversibility of light, we designed the structural colour pixels at this focal plane, so that transmitted light from the pixels

would emerge as parallel rays into the far field. Our microlens showed spherical aberration and chromatic aberration. In geometric optics[35], spherical aberration is caused by the spherical microlens which focuses marginal rays and paraxial rays at different points along the optical axis, whereas chromatic aberration is caused by the dispersion of the material due to its wavelength-dependent refractive index. At $\theta = 16°$ (Fig. 2b), we found that the focal plane shifted in the negative z-direction due to another aberration known as field curvature, which causes light rays with different field angles ($\theta \geq 0°$) to focus on a curved surface rather than on a flat plane. Despite the presence of aberrations, the microlens behaves as intended: an optical router[31] that directs light rays from selected pixel positions to the observer. In the thin lens approximation, the pixel's position selected by the microlens is expressed in Eq. 2:

$$x = F \tan \theta \qquad (2)$$

For a given field angle $\theta$, the microlens forms a focal spot at position $x$. As a change in $\theta$ leads to a change in $x$, the set of light rays focused by the microlens translates across different pixel positions. This translation mechanism enables the microlens to steer the direction of light rays and display changing perspectives of a 3D image at their corresponding viewpoints.

**Electromagnetic wave simulations.** To evaluate the microlens focal spot size, which determined the smallest pixel pitch in our LFP, we simulated the electric field intensity distributions of the focal spot on the plane $z = 37$ μm, for field angle $\theta = 0°$ and wavelengths $\lambda = \{450, 532, 635\}$ nm. We then calculated the full width at half maximum (FWHM) of the electric field intensity profile of the horizontal slice at $y = 0$ (Fig. 2c). For the respective wavelengths, the FWHM = $\{0.85, 1.00, 1.19\}$ μm, which are in close agreement with their own diffraction-limited value (FWHM′) expressed in Eq. 3.

$$\text{FWHM}' = 0.51 \frac{\lambda}{\text{NA}} \qquad (3)$$

where $\lambda$ is the wavelength of light, and NA is the microlens NA. On the same plane $z = 37$ μm, we also simulated how the FWHM changed from $\theta = 0$–16° for the respective wavelengths. The plot of FWHM vs. $\theta$ revealed an overall increasing trend (Fig. 2d), which suggests that image defocus becomes more severe at larger field angles. This trend is expected for a single lens element that is uncorrected for aberrations.

Based on the simulation results of our microlens, we designed our initial pixel pitch ($P = 5$ μm) to be larger than the FWHM. Our initial pixel comprised $5 \times 5$ nanopillars, in which each nanopillar has diameter $D$, height $H$ and pitch $S$ (Fig. 1c). Though these nanopillars have previously been shown to produce a wide range of colours[8,30,31], their colour change with viewing angle has not yet been characterized. To simulate a representative pixel, we set $D = 0.3$ μm, $H = 1.2$ μm, $S = 1$ μm, and calculated its transmittance spectra for different viewing angles $\phi$ (Fig. 2e). From $\phi = 0°$ to 8°, the shape of the spectra remained unchanged. However, from $\phi = 8°$ to 16°, the trough of the spectra shifted from 570 to 586 nm and increased its transmittance from 3 to 12%, which suggests that the spectral contrast and saturation are reduced at larger viewing angles. Hence, the NA of our microlens (NA ~0.28) was optimal for our structural colour pixels, as it covered an acceptable range of viewing angles from $\phi = 0°$ to 16°. Within this range of angles, we expect to see minimal colour change in any pixel, or in the image of the LFP.

**Fabrication and observation.** We fabricated our LFP by TPL of IP-Dip photoresist on a glass substrate. To create a wide range of

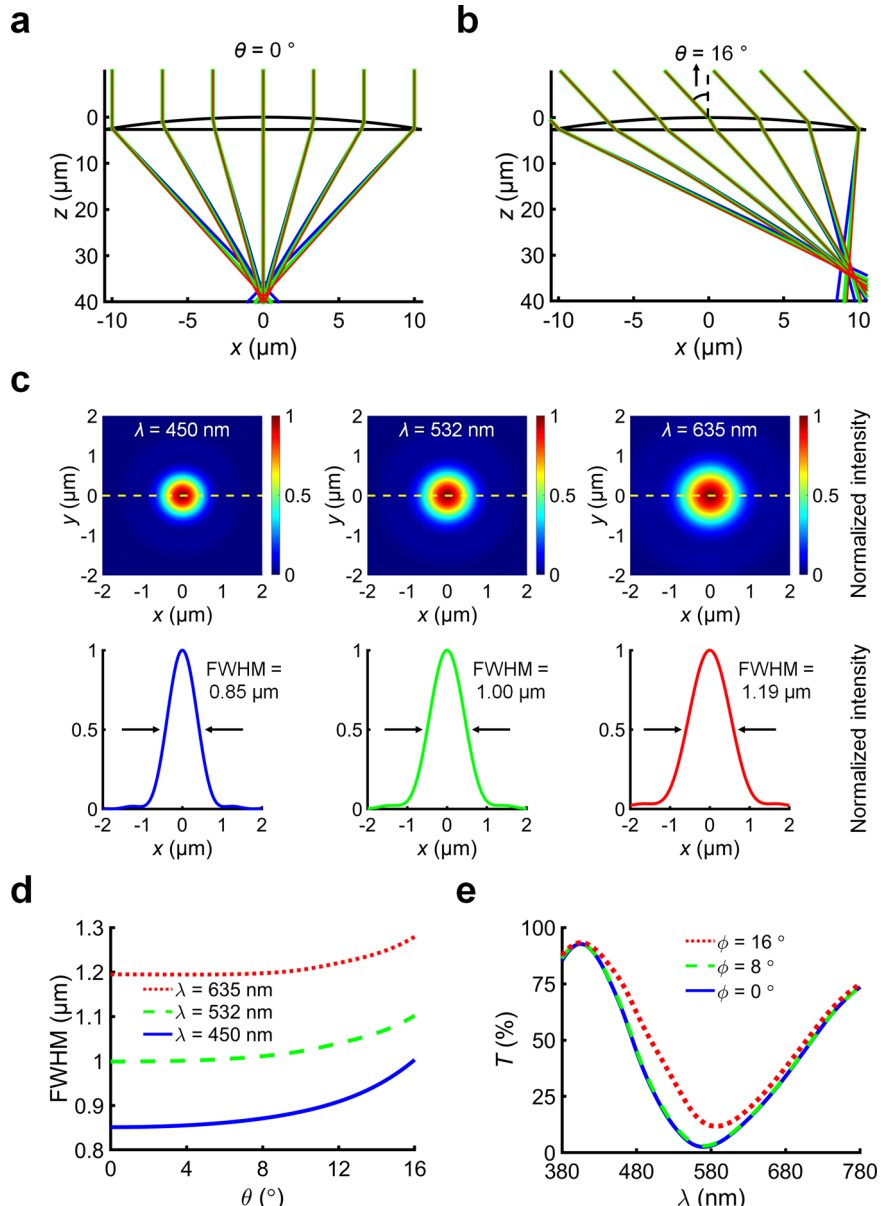

**Fig. 2 Numerical simulations of the microlens and colour pixel. a, b** Raytracing simulation of the microlens for field angles $\theta = \{0, 16\}°$ that select the centre and corner pixel positions, respectively. **c** Simulated normalized electric field intensity distribution of the $x$-$y$ focal plane at $z = 37\ \mu m$, for $\theta = 0°$ and wavelengths $\lambda = \{450, 532, 635\}$ nm, respectively. The focal spot size was calculated as the full width at half maximum (FWHM) of the normalized electric field intensity profile of the horizontal slice $y = 0$, indicated by the yellow dashed line. **d** Plot of the simulated FWHM vs. $\theta$ for the respective wavelengths. **e** Simulated transmittance spectra of a representative pixel (which comprised $5 \times 5$ nanopillars with diameter $D = 0.3\ \mu m$, height $H = 1.2\ \mu m$, pitch $S = 1\ \mu m$) for viewing angles $\phi = \{0, 8, 16\}°$.

structural colour pixels, we varied the height of nanopillars from 0.6 and 2.7 μm in steps of 0.3 μm, and the laser exposure time from 0.04 to 0.32 ms in steps of 0.04 ms per exposed voxel. The laser exposure time controlled the diameter of nanopillars. All nanopillars were exposed by femtosecond laser pulses, with the laser power set to 20 mW. We then measured the transmittance spectra of the structural colour pixels and plotted their coordinates on the CIE chromaticity diagram (Supplementary Fig. 1 and Supplementary Note 2). We used the CIE data to map the structural colour pixels of our LFP to their patterning parameters. To reduce printing time, the laser scanned only the surface shell of the towers and microlenses. Uncured photoresist within the shell was later solidified by UV curing[36]. The display units were fabricated in a pseudorandom arrangement, such that the centre-to-centre separation between adjacent display units ranged from

29 to 45 μm in $x$ and $y$ directions. Under an optical microscope, the display units of our LFP looked like colourful compound eyes as found in insects (Fig. 3a). Though each display unit comprised a microlens and a block of pixels with multiple colours, a bright single-colour spot was produced when the optical microscope was focused at the front focal plane of the display unit. This bright single-colour spot was caused by the mixing of different wavelengths of light from pixels under each microlens (Supplementary Fig. 2).

To demonstrate the colour mixing effect, we fabricated a display unit and a block of pixels next to each other on a separate substrate. We designed the pixels with equal areas of red, blue, and green stripes. The same pixels were patterned beneath the microlens of the display unit. When the optical microscope was focused on the pixels, the stripes were clearly resolved (Fig. 3b).

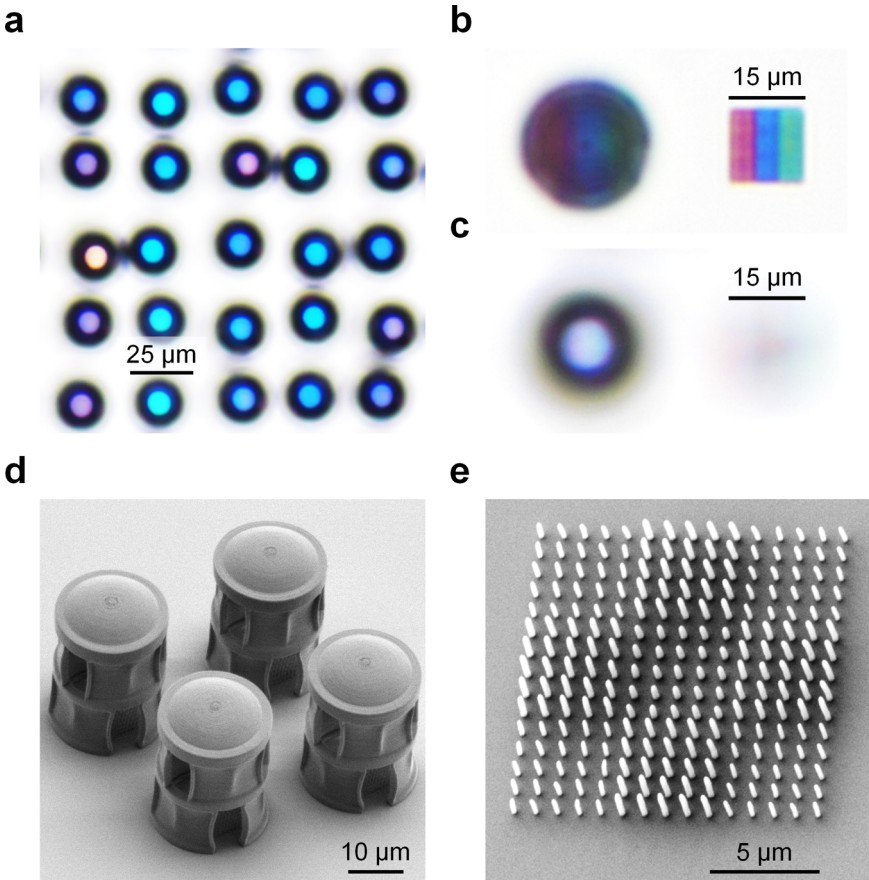

**Fig. 3 Optical and electron micrographs of the light field print. a** Brightfield transmission optical microscope image (plan-view) of the display units in a pseudorandom arrangement. **b**, **c** Brightfield transmission optical microscope images of a display unit (left) and a block of structural colour pixels designed with red, blue, and green stripes (right). The same pixels were patterned beneath the microlens of the display unit. **b** The optical microscope image was focused on the pixels. **c** The optical microscope image was focused on the front focal plane of the display unit. **d** Scanning electron microscope (SEM) image (45° tilt angle) of 2 × 2 display units. **e** SEM image (30° tilt angle) of 3 × 3 pixels, in which each pixel is recognizable by 5 × 5 nanopillars with the same height and diameter.

However, when the optical microscope was focused on the front focal plane of the display unit, the stripes became out-of-focus, and a bright white spot was formed above the display unit (Fig. 3c). The spot appeared white because the stripes had equal areas, so the colour of each stripe contributed in equal proportion to the total spectral power distribution of the spot. Hence, larger areas of pixels with the same colour will contribute a larger proportion to the total spectral power distribution and influence the resultant colour of the spot. We also fabricated 2 × 2 display units to examine their structure under a scanning electron microscope (SEM) (Fig. 3d). As the microlenses were printed at a discrete slicing height step of 20 nm, their layering artefacts are visible in the SEM image. Despite the presence of layering artefacts, the microlenses still showed good optical performance. To test the microlens performance, we used a laser setup to capture focal spot images for central wavelengths $\lambda = \{450, 532, 635\}$ nm (Supplementary Fig. 3). In each focal spot image, we measured the intensity distribution of its centre horizontal slice, then calculated the FWHM = $\{0.88, 1.03, 1.21\}$ μm and Strehl ratio = $\{0.88, 0.88, 0.89\}$ for the respective wavelengths. The Strehl ratio was calculated by normalizing the intensity distribution of the horizontal slice with the same area under the curve as an ideal Airy disk function[37]. A Strehl ratio >0.8 suggests that the microlens performance approaches the diffraction limit. We fabricated another 3 × 3 structural colour pixels to examine their nanopillars under the SEM (Fig. 3e). Each pixel comprised 5 × 5

nanopillars with the same height and diameter. The pixels are distinguishable from one another in the SEM image, as adjacent pixels have different heights and diameters of nanopillars. Due to the similar structure of nanopillars in each pixel, its colour appeared uniform under the optical microscope.

Next, we used a white light-emitting diode lamp to illuminate our LFP and observed it from several viewpoints in the far field. The viewpoints are indicated by azimuthal angle $\alpha = \{-8, 0, 8\}°$ and elevation angle $\beta = \{-8, 0, 8\}°$ relative to the centre normal line of sight (Fig. 1a). We captured digital camera images of the LFP, which revealed different perspectives of the cube at different viewpoints (Fig. 4). These images have a grainy appearance as they were captured by the camera's macro lens that resolved individual microlenses in the LFP. However, the LFP appeared unpixelated to the naked eye as the centre-to-centre separation between microlenses (29–45 μm) was smaller than the eye's resolving power (~50 μm). We observed that the colours of the cube were consistent throughout the acceptable range of viewing angles ($\phi = 0–16°$), which agreed with our simulation results in Fig. 2e. When the camera was placed in between the specified viewpoints, we observed an image with different perspectives of the cube fused together, which we refer to as crosstalk (Supplementary Fig. 4). Crosstalk was caused by the position of the microlens focal spot on at least two adjacent structural colour pixels in each display unit. To avoid crosstalk, the focal spot should be smaller than one pixel and positioned within the area of

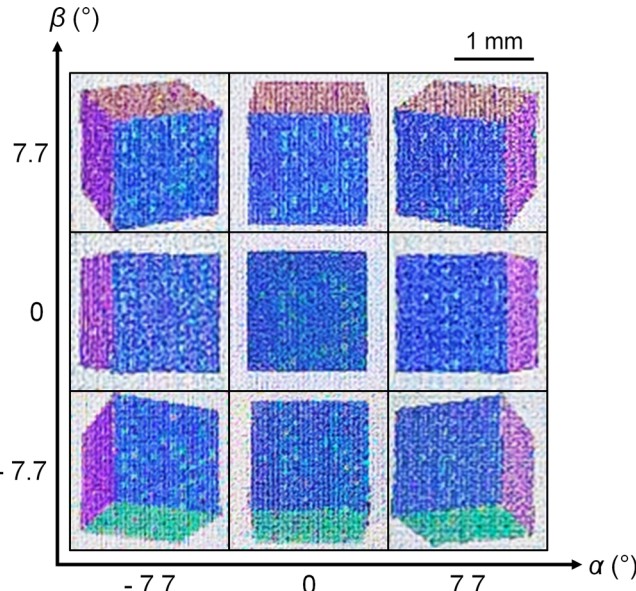

**Fig. 4 Light field print encoded with 3 × 3 perspectives of a multi-coloured cube.** Digital camera macro images of the cube were captured from each viewpoint denoted by azimuthal angle $\alpha$ and elevation angle $\beta$. The cube appeared to protrude out of the plane of the substrate. In this light field print, each colour pixel comprised 5 × 5 nanopillars (pixel pitch $P = 5\,\mu m$). The same scale bar applies to each viewpoint image.

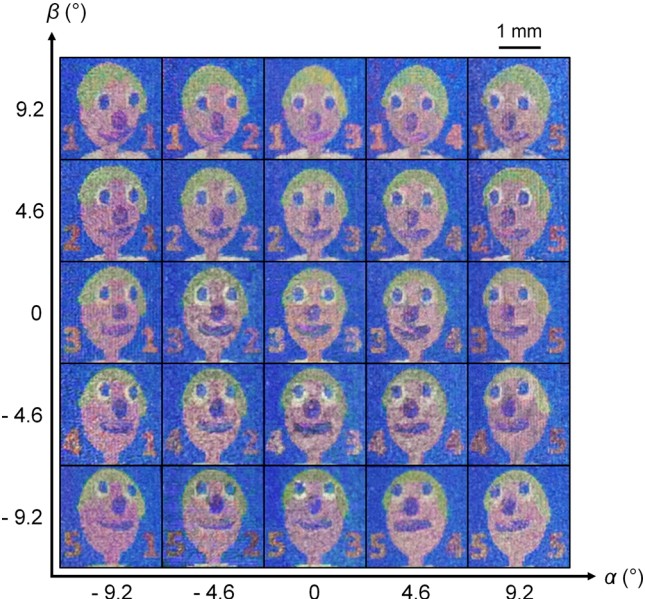

**Fig. 5 Light field print encoded with 5 × 5 perspectives of a computer-generated cartoon face.** Digital camera macro images of the cartoon face were captured from each viewpoint denoted by azimuthal angle $\alpha$ and elevation angle $\beta$. The cartoon face appeared to protrude out of the plane of the substrate. In this light field print, each colour pixel comprised 3 × 3 nanopillars (pixel pitch $P = 3\,\mu m$). The same scale bar applies to each viewpoint image.

any selected pixel. As each pixel encoded a discrete perspective of the 3D object, the image transition between viewpoints was quasi-continuous, which gave an impression of motion parallax. Smoother motion parallax can be achieved by reducing pixel pitch and encoding the LFP with more input images that have smaller angular differences in perspective.

**Effects of reducing pixel pitch.** To investigate how smaller pixel pitch affects image crosstalk and the smoothness of motion parallax, we fabricated another LFP in which each pixel comprised 3 × 3 nanopillars. The pitch between nanopillars remained as $S = 1\,\mu m$, but the pixel pitch was reduced to $P = 3\,\mu m$. In designing the LFP, we used an open-source 3D model of a cartoon face (http://objects.povworld.org/cgi-bin/dl.cgi?soldier.zip) and computer graphics software (POV-Ray) to generate 5 × 5 input images with different perspectives. We encoded each input image with a row number on the left side, and a column number on the right side. The row number increases from top to bottom viewpoints, and the column number increases from left to right viewpoints. These numbers served as visual cues for us to position the camera at each specified viewpoint and avoid crosstalk when capturing images of the LFP. We followed the same design and fabrication process, and captured images of the LFP from different viewpoints (Fig. 5). Due to the larger number of input images, we found that these captured images of the cartoon face ($P = 3\,\mu m$) showed crosstalk at more transition viewpoints than the images of the cube ($P = 5\,\mu m$). However, the captured images of the cartoon face also showed smoother motion parallax than the images of the cube due to a smaller angular sampling interval of the LFP. The angular sampling interval $\omega_a$ is expressed in Eq. 4:

$$\omega_a = \tan^{-1}\left(\frac{P}{F}\right) \qquad (4)$$

where $P$ is the pixel pitch, and $F$ is the microlens focal length. A smaller value of $\omega_a$ indicates higher angular resolution and smoother motion parallax.

To determine the highest angular resolution of our LFP, we fabricated each pixel with only a single nanopillar (~300 nm diameter), which was the smallest printable feature size in our TPL system. This LFP was encoded with 15 × 15 input images of the cartoon face and its pixel pitch was $P = 1\,\mu m$, same as the pitch between nanopillars ($S = 1\,\mu m$). The area of the LFP spanned 2 mm × 2 mm. Remarkably, we still observed clear and colourful images with smooth motion parallax, as this result shows that a single nanopillar suffices to represent a colour pixel. We found that reducing the pixel size from 5 × 5 nanopillars to only a single nanopillar had little effect on the appearance of pixel colour and contrast (Supplementary Fig. 5). To achieve the smoothest motion parallax, the angular sampling interval ($\omega_a$) and the angular difference in perspective between input images ($\omega_b$) should be smaller than a threshold angle[38] expressed in Eq. 5:

$$\delta = \tan^{-1}\left(\frac{E}{V}\right) \qquad (5)$$

where $\delta$ is the angle between two sampled light rays that enter the eye, $E$ is the pupil diameter of the eye, and $V$ is the viewing distance from the eye to the 3D image. If this condition is satisfied ($\omega_a \leq \delta$ and $\omega_b \leq \delta$), the eyes will perceive the smoothest motion parallax and resolve the accommodation-vergence conflict[38–40] (i.e. the experience of visual discomfort when the eyes focus on a different point from where they converge). For $E = 6\,mm$ and $V = 150\,mm$ (the close focusing distance of the naked eye), $\delta = 2.3°$. Under close inspection, we observed images with smooth motion parallax because $\omega_a$ (~1.6°) and $\omega_b$ (~2.1°) were smaller than $\delta$. We also observed how the images transitioned smoothly by tilting the substrate horizontally and vertically, while fixing the position of the camera and light source (Supplementary Movie 1). As the substrate was tilted, the varying incident angle of light on the LFP caused variation in its image colour brightness and contrast. The image became white or translucent when the

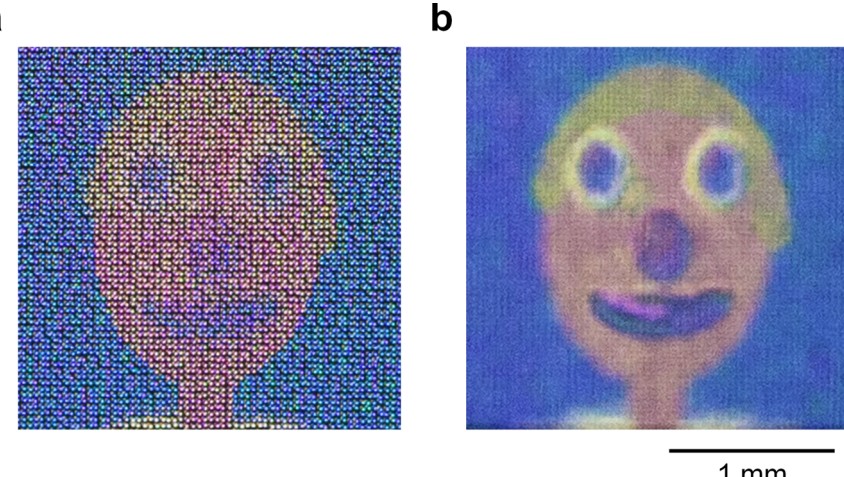

**Fig. 6 Appearance of the light field print captured at different focus distances.** Digital camera macro images of the cartoon face were captured straight on from the central viewpoint (azimuthal angle $\alpha = 0°$ and elevation angle $\beta = 0°$). This light field print was encoded with $15 \times 15$ different perspectives of the cartoon face, and each colour pixel comprised only a single nanopillar ~300 nm diameter (pixel pitch $P = 1\,\mu m$). **a** Image captured with the camera focused on the plane of the microlenses. **b** Image captured with the camera focused slightly in front of the microlenses, which also simulates what an observer sees. The same scale bar applies to both images.

substrate was tilted beyond the largest acceptable viewing angle ~16°, which caused the microlenses to select an area with no pixels on the substrate. We verified that the LFP formed a 3D image when we focused the camera on the microlenses (Fig. 6a) and slightly above the microlenses (Fig. 6b). The image appeared clearer in the latter case, which was closer to what we observed by naked eye. The images in Fig. 6 revealed the pixelated composition of the LFP as they were captured by the macro lens of the camera that resolved individual microlenses in the LFP. To mitigate the pixelated appearance of the image, the LFP can be fabricated with a larger total area and captured at reduced magnification. By using a simplified geometric model[41] (Supplementary Note 3), we calculated the maximum image depth to be 1.3 mm, which refers to how much the 3D image appeared to float above or sink below the LFP as seen from the centre viewpoint. The maximum image depth $I$ is expressed in Eq. 6:

$$I = \frac{CF}{P} \qquad (6)$$

where $C$ is the centre-to-centre separation distance of the microlens, $F$ is the focal length of the microlens, and $P$ is the pixel pitch. In the calculation of maximum image depth, we used an average $C$ of 37 $\mu m$. Hence, the total depth range of the 3D image was $1.3\,mm \times 2 = 2.6\,mm$. Our LFP worked in transmission mode, where the light source and observer are on opposite sides of the glass substrate. The LFP can also work in reflection mode by adding a mirror on the clean side of the substrate and placing the light source on the same side as the observer. However, the image brightness and contrast were reduced in reflection mode (Supplementary Fig. 6) due to more light scattering and optical losses at the interfaces of the microlenses, which gave the LFP a washed-out appearance.

## Discussion

We have presented a nanoscale 3D printing approach to directly fabricate high-resolution LFPs by TPL. Our fabrication process only involved patterning IP-Dip photoresist on a glass substrate and developing the photoresist in chemical solution, along with UV curing. We fabricated plano-convex spherical microlenses with NA ~0.28 and structural colour pixels with nanopillars, that were designed to display directional information across a range of viewing angles from 0° to 16°. We systematically reduced the size

of each pixel from $5 \times 5$ nanopillars to just a single nanopillar (~300 nm in diameter) and found that it still produced clear and colourful images with smooth motion parallax. By fabricating single nanopillar pixels in our LFP, we achieved a maximum pixel resolution of 25,400 dots per inch. To the best of our knowledge, this pixel resolution is the highest for LFPs that have been reported so far. From the palette of 64 colours (6 bits) in Supplementary Fig. 1a, we calculated the maximum amount of information stored in our LFP to be 0.98 Megabits per mm². We also achieved a LFP with simultaneously high spatial resolution (29–45 $\mu m$) and high angular resolution (~1.6°). Ultimately, the highest spatial resolution is determined by the smallest centre-to-centre separation between microlenses, and the highest angular resolution is determined by the maximum density of pixels under the microlenses.

Due to the serial patterning process of our TPL system, it took about 24 h to print our LFP with an area that spanned $2\,mm \times 2\,mm$. Though this process currently limits upscaling and mass production, it can be improved by using parallel processing TPL systems[42,43] that increase the throughput by several orders of magnitude while maintaining sub-micron features and accurate alignment between the microlenses and pixels. Sub-micron features are needed to create a high-resolution LFP, whereas accurate alignment is essential to reconstruct a clear 3D image. The microlenses and pixels of our LFP were aligned automatically in the TPL system, which eliminated the need for doing manual alignment. The fabrication process can be further optimized by using more sensitive photoresists, higher laser powers and improving the design of microlenses and pixels. As the total area of the LFP is scaled up, the design needs to consider optimal observer positions based on the NA of microlenses to avoid an optical problem where some parts of the image lie outside the acceptable range of viewing angles. The NA of a microlens is determined by its diameter and focal length. In our LFP, we set the focal length equal to the separation distance between microlenses and pixels. This separation distance also needs to be carefully designed. A larger separation distance increases the aspect ratio of the support structure, which makes it less stable and less feasible to fabricate. We suggest that the aspect ratio should not exceed ~10:1, which was the case in our LFP. Conversely, a smaller separation distance would produce less saturated colours as unwanted wavelengths of light scattered from the nanopillar pixels are collected by microlenses with larger NA. Hence, the separation distance has an upper limit

determined by the mechanical stability of high aspect ratio support structures and the minimum acceptable range of viewing angles. The lower limit is determined by matching the NA of microlens with the maximum viewing angle of pixels (Fig. 2e) to avoid reducing colour saturation.

In terms of security, copying the LFP by nanoimprint lithography would be extremely difficult because the lenses shield the pixels from being moulded, which can provide a physically unclonable function for anti-counterfeiting. At the same time, the LFP is easy to observe by naked eye under ambient white light illumination and its optically variable feature can benefit security documents, such as passports. The LFP does not require special glasses for viewing, nor does it require laser illumination used in conventional holograms. In terms of performance, the pixels of our LFP covered only a limited colour gamut (~40% sRGB) as they were made entirely of IP-Dip photoresist, a low refractive index material. By contrast, pixels made of high refractive index materials, such as silicon, can yield a much wider colour gamut[19,44,45] covering up to the Rec. 2020 colour space. However, high refractive index materials also introduce stronger dispersion and reflective losses that diminish the optical performance of microlenses. Whilst more research is needed in nanoscale 3D printing of high refractive index materials[46], the choice of a low refractive index material allowed the microlenses and pixels to be printed in a single process that greatly reduced the design constraints of our LFP. Another limitation is that the fabricated structures of the LFP are fragile and can be easily wiped off by hand. To protect the LFP from structural damage, we suggest keeping it inside a glass enclosure under room temperature and pressure. Nonetheless, we emphasize the main advantage of nanoscale 3D printing in creating not only high-resolution, but also fully customizable LFPs.

In the future, hyper-realistic LFPs can be created by integrating high optical performance metalenses[47–50] and structural colour pixels. Hyper-realistic LFPs will require a wide range of viewing angles to deliver an immersive 3D experience. This goal can be achieved by sophisticated design and fabrication of metalenses[51,52] that provide large fields of view up to 180° and correct for field-dependent aberrations. To reduce unwanted iridescence over a wide range of viewing angles, angle-insensitive structural colour pixels[53,54] should also be designed and fabricated. In addition, the metalenses and structural colour pixels can be fabricated on curved substrates[55] instead of flat substrates. If high NA metalenses (NA ~1) are combined with angle-insensitive structural colour pixels of 250 nm pitch, a hyper-realistic LFP that provides a ~180° range of viewing angles and an angular resolution of 0.4° can be achieved. Hyper-realistic LFPs will find potential applications in artworks and product protection features aimed at the high-value market.

## Methods

**Simulations.** Raytracing simulation of the microlens was done in MATLAB by defining the geometric profile of the microlens and applying Snell's law at each interface (Supplementary Note 1). Electromagnetic wave simulation of the microlens and structural colour pixel was done in Lumerical FDTD Solutions by using the finite-difference time-domain method. The microlens surface was defined by diameter = 21 µm, radius of curvature = 22 µm, conic constant = 0. A plane wave source was launched from the convex side of the microlens and its electric field intensity distribution at the focal plane ($z = 37$ µm) was recorded and normalized to the maximum intensity. The structural colour pixel contained an array of $5 \times 5$ nanopillars on a glass substrate. Each nanopillar was modelled as a cylinder capped with a hemisphere, with diameter = 0.3 µm, height = 1.2 µm and pitch = 1 µm, which were average values measured from the SEM images. A total-field scattered-field source was launched from the substrate side. The power transmitted into the far field was calculated within a 16° integration cone. The centre of the integration cone was tilted at 0°, 8°, and 16° to simulate different viewing angles. The calculated spectra were normalized to a reference transmission spectrum from a control setup without the nanopillars.

**Fabrication.** A drop of IP-Dip photoresist was placed on a fused silica substrate and patterned by a Nanoscribe GmbH Photonic Professional GT system. The writing mode was set to GalvoScanMode, and the laser's power was set to 20 mW. The nanopillars of the pixels were written in PulsedMode, while the towers and microlenses were written in ContinuousMode. After patterning, the substrate was immersed in propylene glycol monomethyl ether acetate solution for 11 min, then in isopropyl alcohol solution for 2 min with UV curing, and then in nonafluorobutyl methyl ether solution for 5 min. Lastly, the substrate was dried in air.

**Imaging.** The optical microscope images in Fig. 3a–c were taken with a Nikon T Plan EPI SLWD 10× NA0.2 objective attached to a Nikon DS-Ri2 microscope camera. The SEM images in Fig. 3d were taken with a JEOL JSM-7600F field emission SEM system. The digital camera images in Figs. 4–6 were taken with a Canon EF 100 mm f/2.8 Macro USM lens mounted on a Canon EOS 5D Mark III DSLR camera set to manual exposure mode: ISO 800, f/4, 1/100s. The colour brightness and contrast of the digital camera images were adjusted via image processing to provide a consistent appearance for presentation.

## Data availability

The data that support the findings of this work are available from the corresponding authors upon reasonable request.

## Code availability

The code used to fabricate the light field print is available from the corresponding authors upon reasonable request.

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

## Acknowledgements

This research is supported by National Research Foundation (NRF) Singapore, under its Competitive Research Programme award NRF-CRP20-2017-0004 and NRF Investigatorship Award NRF-NRFI06-2020-0005. J.Y.E.C. thanks Andrew Bettiol, Jay Teo and Yek Soon Lok for their mentorship.

## Author contributions

J.Y.E.C. did the experiments and simulations, captured the images, analysed the results, and wrote the manuscript. Q.F.R. assisted in the experiments, simulations, and analysis of results. M.H.J., H.T.W., H.W. and W.Z. assisted in the analysis of results and drawing of schematics. C.W.Q. and J.K.W.Y. supervised the research. All authors edited the manuscript.

## Competing interests

The authors declare no competing interests.
