## [Peer Review File · Nature Communications]

REVIEWER COMMENTS

Reviewer #1 (Remarks to the Author):

This is an interesting and carefully conducted research exercise involving innovative precision engineering, simulation, and the demonstration of proof of concept. As the researchers indicate, the work fundamentally builds on the 3D display technique pioneered by Lippmann at the beginning of the twentieth century. The techniques devised by the researchers find solutions to longstanding difficulties associated with taking Lippmann's work forward in the development of a precision lightfield display. This includes the development of a one-step technique for the implementation of the optical arrangement coupled with ensuring accurate alignment of the lenslet array with the pixel array. When coupled with the ability of the prototype system to demonstrate sound spatial and angular resolutions, it is evident that this fabrication/optimisation work may well provide a significant advance in 3D imaging.

Assuming that the small prototype systems are scalable in terms of physical display dimensions, then it would seem that there are a broad range of diverse applications which could benefit from this methodology. The researchers indicate (circa line 433) that the fabrication of a 2x2 square millimetre area occupied twenty-four hours, but suggest that a parallel projection TPL system could increase throughput by several orders of magnitude. If this paper is to be accepted for publication by Nature I would strongly suggest that this statement requires further clarification. Moving from an area of 4 square millimetres to 400 square centimetres may be feasible in terms of parallel printing but I would suggest that as a result, optical problems are likely to be encountered. This needs some careful consideration and discussion. Also, as a minor point, why are the linear dimensions indicated as being square millimetres?

Also, I feel that text circa line 384 should be clarified - specifically the indication that 'remarkably, we still observed colourful images with smooth motion parallax'. Whilst I discern the researchers' meaning, what was the size of the display panel under observation?

The fabrication of the lenslets is a crucial aspect of this work. However, unless I have missed some point, the discrete nature of the slicing heights does not appear to be mentioned until line 294. In terms of readability of this publication, this should be clarified early in the paper, otherwise a reader who does not specialise in this production technique may be disadvantaged.

Assuming that the above points are addressed, I would recommend this work for publication in Nature.

Barry G Blundell

Reviewer #2 (Remarks to the Author):

The paper High-resolution light field prints by nanoscale 3D printing brings new technology to the important, yet old topic of light field print. Basically, the challenge is to describe 3D information in a 2D or semi 2D manner. One way to achieve this goal is by superposing many sub images that are slightly shifted with respect to one another, e.g. by the use of lens array that is placed in close proximity to the image. The importance of this field comes from the diverse potential applications, be it either art or holograms for security. The main challenge is to achieve high quality prints, with high spatial and angular resolution. Mitigating these goals using conventional technologies is extremely challenging as inkjet printing fails to achieve high resolution prints. The more recent use of plasmonic printing combined with an array of micro lenses also fails to achieve satisfactory results, mostly due to misalignment between the print and the micro lens array. Here comes the main contribution of this paper. Using advanced nanofabrication tools in 3D, the authors demonstrated high resolution prints that are combined with micro lens array in the same nanofabrication process to achieve high quality printing that are of sufficient resolution to the

naked eye, and are practically free of misalignment. While the main achievement is in the technology rather in the basic science, I see this result as interesting, important, and of high impact, going significantly beyond the current state of the art in the field. As such, I recommend publishing this paper in nature communications. Before publication, I would like the authors to address the following issues:

1 - The current demonstration is based on the use of 3-D printing by 2 photon polymerization of a photoresist. This dictates a rather low refractive index of the metasurface that is used to generate color. I urge the authors to discuss this issue, address the limitation and project what could be improved if higher refractive index can be used.

2 - A major contribution is the integration of microlens array and color structures in the same process. what are the limitations in terms of dimension (e.g. separation between the lens and the print), and how does it affect the performance.

3 - Following the previous comments, the current NA is ~ 0.28 , which matches the viewing angles for which the colors are almost constant with the angle. However, I guess that there is a need in the future for larger angular bandwidth, which can be increased by both increasing the viewing angle and the angular resolution. What is the proposed strategy to go along this line? a discussion of this aspect could benefit the paper.

4 - the current demonstration is based on light transmission as the source and the viewer are located on the opposite sides of the samples. However, in many realistic scenarios, a reflective type sample is desired. Could the authors discuss the main differences between transmission and reflection mode and suggest a strategy for reflection?

5 - Cost effective manufacturability- this should be a major goal of the technology. As of now, the print is achieved one by one using a fairly complicated Nanoscribe system. How long does it take to print a sample, and what would be the best way to push this technology into cost effective mass production?

6 - Looking at Fig 6b (which is more or less what the observer sees), I can still observe the pixelated structure. How can this be removed, such that the picture will be more smooth?

7 - for encoding purposes, the amount of information is extremely important. What could be the utmost amount of information that can be encoded per unit area?

8 - How stable is the fabricated structure? is it mechanically solid? what is the range of temperatures for which the image is practically unchanged?

9 - the author tested the highest angular resolution by printing a single pillar per pixel. What is the effect of on the quality of the color (contrast, etc.)

Response to Reviewer #1

Comments	This is an interesting and carefully conducted research exercise involving innovative precision engineering, simulation, and the demonstration of proof of concept. As the researchers indicate, the work fundamentally builds on the 3D display technique pioneered by Lippmann at the beginning of the twentieth century. The techniques devised by the researchers find solutions to longstanding difficulties associated with taking Lippmann's work forward in the development of a precision lightfield display. This includes the development of a one-step technique for the implementation of the optical arrangement coupled with ensuring accurate alignment of the lenslet array with the pixel array. When coupled with the ability of the prototype system to demonstrate sound spatial and angular resolutions, it is evident that this fabrication/optimisation work may well provide a significant advance in 3D imaging. Assuming that the small prototype systems are scalable in terms of physical display dimensions, then it would seem that there are a broad range of diverse applications which could benefit from this methodology.
Response	We thank the reviewer for providing rigorous comments about this work.
Comments	The researchers indicate (circa line 433) that the fabrication of a 2x2 square millimetre area occupied twenty-four hours, but suggest that a parallel projection TPL system could increase throughput by several orders of magnitude. If this paper is to be accepted for publication by Nature I would strongly suggest that this statement requires further clarification. Moving from an area of 4 square millimetres to 400 square centimetres may be feasible in terms of parallel printing but I would suggest that as a result, optical problems are likely to be encountered. This needs some careful consideration and discussion.
Response	The reviewer has a valid concern about optical problems that may arise from parallel printing. We have added some discussion (highlighted yellow) to address this concern. “... Due to the serial patterning process of our TPL system, it took about 24 hours to print our LFP with an area that spanned 2 mm × 2 mm. Though this process currently limits upscaling and mass production, it can be improved by using parallel processing TPL systems^{42,43} that increase the throughput by several orders of magnitude while maintaining sub-micron features and accurate alignment between the microlenses and pixels. Sub-micron features are needed to create a high-resolution LFP, whereas accurate alignment is essential to reconstruct a clear 3D image. The microlenses and pixels of our LFP were aligned automatically in the TPL system, which eliminated the need for doing manual alignment. The fabrication process can be further optimized by using more sensitive photoresists, higher laser powers and improving the design of microlenses and pixels. As the total area of the LFP is scaled up, the design needs to consider optimal observer positions based on the NA of microlenses to avoid

	an optical problem where some parts of the image lie outside the acceptable range of viewing angles.” Kindly note that we have included Nat. Commun. 10, 2179 (2019) as Ref 43 to support our discussion on high-speed TPL.
Comments	Also, as a minor point, why are the linear dimensions indicated as being square millimetres?
Response	Square millimetres referred to the total area of the print. In the original sentence, we wrote: “Due to the serial patterning process of our TPL system, it took about 24 hours to print an area of $2 \times 2 \text{ mm}^2$”. To avoid ambiguity, we have revised the sentence to: “Due to the serial patterning process of our TPL system, it took about 24 hours to print our LFP with an area that spanned $2 \text{ mm} \times 2 \text{ mm}$.”
Comments	Also, I feel that text circa line 384 should be clarified - specifically the indication that 'remarkably, we still observed colourful images with smooth motion parallax'. Whilst I discern the researchers' meaning, what was the size of the display panel under observation?
Response	We have added some details and explanation (highlighted yellow) to clarify the meaning of the sentence. “... This LFP was encoded with 15×15 input images of the cartoon face and its pixel pitch was $P = 1 \text{ }\mu\text{m}$, same as the pitch between nanopillars ($S = 1 \text{ }\mu\text{m}$). The area of the LFP spanned $2 \text{ mm} \times 2 \text{ mm}$. Remarkably, we still observed clear and colourful images with smooth motion parallax, as this result shows that a single nanopillar suffices to represent a colour pixel.”
Comments	The fabrication of the lenslets is a crucial aspect of this work. However, unless I have missed some point, the discrete nature of the slicing heights does not appear to be mentioned until line 294. In terms of readability of this publication, this should be clarified early in the paper, otherwise a reader who does not specialise in this production technique may be disadvantaged.
Response	We agree with the reviewer that this important point should be clarified early in the paper. We have added the following sentence (highlighted yellow) in the Introduction section: “... The microlenses and structural colour pixels of our LFPs were aligned automatically in the TPL system (Nanoscribe GmbH Photonic Professional GT system), which can position each volumetric pixel exposed by the laser up to an accuracy of 10 nm. As TPL is an additive manufacturing technology, the

	microlenses and structural colour pixels were fabricated in discrete slicing height steps of 20 nm and 300 nm respectively.”
Comments	Assuming that the above points are addressed, I would recommend this work for publication in Nature. Barry G Blundell
Response	We have meticulously taken note of the points raised by the reviewer and addressed them. We sincerely thank the reviewer for giving this work a positive recommendation.

Response to Reviewer #2

Comments	The paper High-resolution light field prints by nanoscale 3D printing brings new technology to the important, yet old topic of light field print. Basically, the challenge is to describe 3D information in a 2D or semi 2D manner. One way to achieve this goal is by superposing many sub images that are slightly shifted with respect to one another, e.g. by the use of lens array that is placed in close proximity to the image. The importance of this field comes from the diverse potential applications, be it either art or holograms for security. The main challenge is to achieve high quality prints, with high spatial and angular resolution. Mitigating these goals using conventional technologies is extremely challenging as inkjet printing fails to achieve high resolution prints. The more recent use of plasmonic printing combined with an array of micro lenses also fails to achieve satisfactory results, mostly due to misalignment between the print and the micro lens array. Here comes the main contribution of this paper. Using advanced nanofabrication tools in 3D, the authors demonstrated high resolution prints that are combined with micro lens array in the same nanofabrication process to achieve high quality printing that are of sufficient resolution to the naked eye, and are practically free of misalignment. While the main achievement is in the technology rather in the basic science, I see this result and interesting, important, and of high impact, going significantly beyond the current state of the art in the field. As such, I recommend publishing this paper in nature communications. Before publication, I would like the authors to address the following issues:
Response	We sincerely thank the reviewer for providing rigorous comments about this work and giving it a positive recommendation. We have meticulously taken note of the points below raised by the reviewer and addressed them.
Comments	1 - The current demonstration is based on the use of 3-D printing by 2 photon polymerization of a photoresist. This dictates a rather low refractive index of the metasurface that is used to generate color. I urge the authors to discuss this issue, address the limitation and project what could be improved if higher refractive index can be used.
Response	We agree with the reviewer that there are limitations, but also advantages, of using low refractive index materials. We have added some discussion (highlighted yellow) to address this issue. “... The LFP does not require special glasses for viewing, nor does it require laser illumination used in conventional holograms. In terms of performance, the pixels of our LFP covered only a limited colour gamut (~ 40% sRGB) as they were made entirely of IP-Dip photoresist, a low refractive index material. By contrast, pixels made of high refractive index materials such as silicon can yield a much wider colour gamut^{19,44,45} covering up to the Rec. 2020 colour space. However, the downside is that high refractive index materials also introduce

	stronger dispersion and reflective losses that diminish the optical performance of microlenses. Whilst more research is needed in nanoscale 3D printing of high refractive index materials⁴⁶, the choice of a low refractive index material allowed the microlenses and pixels to be printed in a single process that greatly reduced the design constraints of our LFP.” Kindly note that we have included Nano Lett. 20, 3513-3520 (2020) as Ref 46 to support our discussion on a report of a high index material (TiO₂) used in nanoscale 3D printing.
Comments	2 - A major contribution is the integration of microlens array and color structures in the same process. what are the limitations in terms of dimension (e.g. separation between the lens and the print), and how does it affect the performance.
Response	We have added some discussion (highlighted yellow) on how the separation distance between microlenses and pixels affects the performance of the light field print. A full discussion on the design of microlenses and pixels will be too much and detract from the main message of this paper. We prefer to write a separate report on these design considerations. “... As the total area of the LFP is scaled up, the design needs to consider optimal observer positions based on the NA of microlenses to avoid an optical problem where some parts of the image lie outside the acceptable range of viewing angles. The NA of a microlens is determined by its diameter and focal length. In our LFP, we set the focal length equal to the separation distance between microlenses and pixels. This separation distance also needs to be carefully designed. A larger separation distance increases the aspect ratio of the support structure, which makes it less stable and less feasible to fabricate. We suggest that the aspect ratio should not exceed ~ 10:1, which was the case in our LFP. Conversely, a smaller separation distance would produce less saturated colours as unwanted wavelengths of light scattered from the nanopillar pixels are collected by microlenses with larger NA. Hence, the separation distance has an upper limit determined by the mechanical stability of high aspect ratio support structures and the minimum acceptable range of viewing angles. The lower limit is determined by matching the NA of microlenses with the maximum viewing angle of pixels (Fig. 2e) to avoid reducing colour saturation.”
Comments	3 - Following the previous comments, the current NA is ~0.28, which matches the viewing angles for which the colors are almost constant with the angle. However, I guess that there is a need in the future for larger angular bandwidth, which can be increased by both increasing the viewing angle and the angular resolution. What is the proposed strategy to go along this line? a discussion of this aspect could benefit the paper.

Response	We agree with the reviewer that larger angular bandwidth is needed for LFPs in the future. We have revised our strategy (highlighted yellow) to increase the range of viewing angles: “... In the future, hyper-realistic LFPs can be created by integrating high optical performance metalenses⁴⁷⁻⁵⁰ and structural colour pixels. Hyper-realistic LFPs will require a wide range of viewing angles to deliver an immersive 3D experience. This goal can be achieved by sophisticated design and fabrication of metalenses^{51,52} that provide large fields of view up to 180 ° and correct for field-dependent aberrations. To reduce unwanted iridescence over a wide range of viewing angles, angle-insensitive structural colour pixels^{53,54} should also be designed and fabricated. In addition, the metalenses and structural colour pixels can be fabricated on curved substrates⁵⁵ instead of flat substrates. If high numerical aperture metalenses ($NA \sim 1$) are combined with angle-insensitive structural colour pixels of 250 nm pitch, a hyper-realistic LFP that provides a ~ 180 ° range of viewing angles and an angular resolution of 0.4 ° can be achieved.” Kindly note that we have included arXiv:2102.07999 as Ref 52 to support our discussion on large field of view metalenses.
Comments	4 - the current demonstration is based on light transmission as the source and the viewer are located on the opposite sides of the samples. However, in many realistic scenarios, a reflective type sample is desired. Could the authors discuss the main differences between transmission and reflection mode and suggest a strategy for reflection?
Response	We have conducted an extra experiment to observe how the light field print behaves in reflection mode and reported the results (highlighted yellow) in the manuscript. “... Hence, the total depth range of the 3D image was $1.3 \text{ mm} \times 2 = 2.6 \text{ mm}$. Our LFP worked in transmission mode, where the light source and observer are on opposite sides of the glass substrate. The LFP can also work in reflection mode by adding a mirror on the clean side of the substrate and placing the light source on the same side as the observer. However, the image brightness and contrast were reduced in reflection mode (Supplementary Fig. 6) due to more light scattering and optical losses at the interfaces of the microlenses, which gave the LFP a washed-out appearance.” We have included the following figure as Supplementary Figure 6 in the Supplementary Information.

Supplementary Figure 6. Appearance of the light field print in reflection mode, where the light source and observer are on the same side as the substrate. **(a)** A matte background was placed beneath the substrate. **(b)** An aluminium mirror was placed beneath the substrate.

Comments	5 - Cost effective manufacturability- this is should be a major goal of the technology. As of now, the print is achieved one by one using a fairly complicated Nanoscribe system. How long does it take to print a sample, and what would be the best way to push this technology into cost effective mass production?
Response	We have revised the discussion (highlighted yellow) on how to push this technology toward mass production. “... Due to the serial patterning process of our TPL system, it took about 24 hours to print our LFP with an area that spanned 2 mm × 2 mm. Though this process currently limits upscaling and mass production, it can be improved by using parallel processing TPL systems^{42,43} that increase the throughput by several orders of magnitude while maintaining sub-micron features and accurate alignment between the microlenses and pixels. Sub-micron features are needed to create a high-resolution LFP, whereas accurate alignment is essential to reconstruct a clear 3D image. The microlenses and pixels of our LFP were aligned automatically in the TPL system, which eliminated the need for doing manual alignment. The fabrication process can be further optimized by using more sensitive photoresists, higher laser powers and improving the design of microlenses and pixels.”
Comments	6 - Looking at Fig 6b (which is more or less what the observer sees), I can still observe the pixilated structure. How can this be removed, such that the picture will be more smooth?
Response	We have added some discussion (highlighted yellow) on the cause of the pixelated appearance and how to mitigate it.

	“... We verified that the LFP formed a 3D image when we focused the camera on the microlenses (Fig. 6a) and slightly above the microlenses (Fig. 6b). The image appeared clearer in the latter case, which was closer to what we observed by naked eye. The images in Fig. 6 revealed the pixelated composition of the LFP as they were captured by the macro lens of the camera that resolved individual microlenses in the LFP. To mitigate the pixelated appearance of the image, the LFP can be fabricated with a larger total area and captured at reduced magnification.” We have also revised the discussion on how to achieve the smoothest motion parallax, as well as the maximum image depth that our LFP can reach. “... To achieve the smoothest motion parallax, we propose that the angular sampling interval (ω_a) and the angular difference in perspective between input images (ω_b) should be smaller than a threshold angle³⁸ expressed in Equation 5: $\delta = \tan^{-1} \left(\frac{E}{V} \right) \quad (5)$ where δ is the angle between two sampled light rays that enter the eye, E is the pupil diameter of the eye, and V is the viewing distance from the eye to the 3D image. If this condition is satisfied (i.e. $\omega_a \leq \delta$ and $\omega_b \leq \delta$), the eyes will perceive the smoothest motion parallax and resolve the accommodation-vergence conflict³⁸⁻⁴⁰ (i.e. the experience of visual discomfort when the eyes focus on a different point from where they converge).” “... By using a simplified geometric model⁴¹ (Supplementary Note 3), we calculated the maximum image depth to be 1.3 mm, which refers to how much the 3D image appeared to float above or sink below the LFP as seen from the centre viewpoint. The maximum image depth I is expressed in Equation 6: $I = \frac{CF}{P} \quad (6)$ where C is the centre-to-centre separation distance of the microlens, F is the focal length of the microlens, and P is the pixel pitch. In the calculation of maximum image depth, we used an average C of 37 μm. Hence, the total depth range of the 3D image was $1.3 \text{ mm} \times 2 = 2.6 \text{ mm}$.”
Comments	7 - for encoding purposes, the amount of information is extremely important. What could be the utmost amount of information that can be encoded per unit area?
Response	We have calculated the maximum amount of information per unit area and added it to the discussion (highlighted yellow). “... By fabricating single nanopillar pixels in our LFP, we achieved a maximum pixel resolution of 25,400 dots per inch. To the best of our knowledge, this pixel resolution is the highest for LFPs that have been reported so far. From the palette

	of 64 colours (6 bits) in Supplementary Fig. 1a, we calculated the maximum amount of information stored in our LFP to be 0.98 Megabits per mm ² .”
Comments	8 - How stable is the fabricated structure? is it mechanically solid? what is the range of temperatures for which the image is practically unchanged?
Response	We have added some discussion (highlighted yellow) to address this concern raised by the reviewer. “... Whilst more research is needed in nanoscale 3D printing of high refractive index materials⁴⁶, the choice of a low refractive index material allowed the microlenses and pixels to be printed in a single process that greatly reduced the design constraints of our LFP. Another limitation is that the fabricated structures of the LFP are fragile and can be easily wiped off by hand. To protect the LFP from structural damage, we suggest keeping it inside a glass enclosure under room temperature and pressure. Nonetheless, we emphasize the main advantage of nanoscale 3D printing in creating not only high-resolution, but also fully customizable LFPs.”
Comments	9 - the author tested the highest angular resolution by printing a single pillar per pixel. What is the effect of on the quality of the color (contrast, etc.)
Response	We have conducted an extra experiment to investigate how smaller pixel sizes affect the quality of colour and reported the results (highlighted yellow) in the manuscript. “... Remarkably, we still observed clear and colourful images with smooth motion parallax, as this result shows that a single nanopillar suffices to represent a colour pixel. We found that reducing the pixel size from 5 × 5 nanopillars to only a single nanopillar had little effect on the appearance of pixel colour and contrast (Supplementary Fig. 5).” We have included the following figure as Supplementary Figure 5 in the Supplementary Information.

Supplementary Figure 5. Brightfield transmission optical microscope images (taken with 50X NA 0.4 objective) showing consistent appearance of pixel colour and contrast as the pixel size is reduced. (a) From left to right, the pixel size is gradually reduced from 5×5 nanopillars to only a single nanopillar. (b) Checkerboard patterns of alternating light and dark colour pixels. From left to right, the pixel size in each checkerboard is reduced from 5×5 nanopillars to 3×3 nanopillars to only a single nanopillar.

REVIEWERS' COMMENTS

Reviewer #1 (Remarks to the Author):

I have reviewed the changes that have been made on the basis of my original review. These address the points that I raised and are most helpful.

I can now recommend this paper for publication in Nature, and wish you well with this very interesting area of research, which has considerable promise.

Dr Barry G Blundell

Reviewer #2 (Remarks to the Author):

The authors have addressed my comments in the most adequate way. I now support the publication of this paper in Nature communications.

Response to Reviewer #1

Comments	I have reviewed the changes that have been made on the basis of my original review. These address the points that I raised and are most helpful. I can now recommend this paper for publication in Nature, and wish you well with this very interesting area of research, which has considerable promise. Dr Barry G Blundell
Response	We thank the reviewer for the time and effort to review this work.

Response to Reviewer #2

Comments	The authors have addressed my comments in the most adequate way. I now support the publication of this paper in Nature communications.
Response	We thank the reviewer for the time and effort to review this work.